# Colorless and Transparent Copolyimides and Their Nanocomposites: Thermo-Optical Properties, Morphologies, and Gas Permeabilities

**DOI:** 10.3390/polym11040585

**Published:** 2019-04-01

**Authors:** Hyeon Il Shin, Young-Je Kwark, Jin-Hae Chang

**Affiliations:** 1Department of Polymer Science and Engineering, Kumoh National Institute of Technology, Gumi 39177, Korea; poweril55@naver.com; 2Department of Organic Materials and Fiber Engineering, Soongsil University, Seoul 06978, Korea; ykwark@ssu.ac.kr

**Keywords:** colorless and transparent copolyimide, organoclay, nanocomposite, film

## Abstract

A series of linear aromatic copolyimides (Co-PIs) were synthesized by reacting 4,4′-biphthalic anhydride (BPA) with various molar contents of 2,2′-bis(trifluoromethyl)benzidine (TFB) and *p*-xylylenediamine (*p*-XDA) in *N*,*N*′-dimethylacetamide (DMAc). Co-PI films were fabricated by solution casting and thermal imidization with poly(amic acid) (PAA) on glass plates. The thermo-optical properties and gas permeabilities of Co-PI films composed of various molar ratios of *p*-XDA (0.2–1.0 relative to BPA) were investigated. Thermal properties were observed to deteriorate with increasing *p*-XDA concentration. However, oxygen-transmission rates (O_2_TRs) and optical transparencies improved with increasing *p*-XDA concentration. Co-PI hybrids with a 1:0.2:0.8 molar ratio of BPA:TFB:*p*-XDA and organically modified hectorite (STN) were prepared by the in situ intercalation method. The morphologies and the thermo-optical and gas permeation properties of the hybrids were examined as functions of STN loading (5–50 wt %). XRD and TEM revealed substantial increases in clay particle agglomeration in the Co-PI hybrid films as the clay loading was increased from 5 to 50 wt %. The coefficient of thermal expansion (CTE) and the O_2_TR of a Co-PI hybrid film were observed to improve with increasing STN concentration; however, its optical transparency decreased gradually with increasing STN concentration.

## 1. Introduction

Colorless and transparent polyimide (CPI) films are usually prepared by reactions of dianhydride and diamine monomers bearing fluorine-containing moieties. These CPI films exhibit good solubilities, thermal stabilities, and optical transparencies that are superior to those of other commercially available PIs [1,2,3]. In general, the low cutoff wavelengths (*λ*_o_) and the colorless nature of these CPI films are attributed to the strong electron-withdrawing groups in their monomeric units that inhibit the formation of charge-transfer complexes (CT-complexes) and decrease intermolecular interactions. However, progressing the fabrication of CPI films largely relies on the design and synthesis of new CPI monomers [4,5].

In order to investigate thermo-optical and gas permeation properties, we designed and synthesized novel 4,4′-biphthalic anhydride-based (BPA-based) copolyimides (Co-PIs) [6,7,8]. A Co-PI typically possesses much lower molecular regularity than the corresponding homopolyimide [9,10]. This decreased regularity leads to fewer intermolecular interactions that, in turn, results in new characteristics, such as modified thermo-optical and gas permeation properties, and solubilities. Furthermore, the properties of Co-PIs can be adjusted by varying the ratio of the dianhydride and diamine comonomers.

Nanocomposites are a class of material that contain ultrafine inorganic particles (with sizes in the nanometer range) that are homogeneously dispersed within polymer matrices. The addition of low concentrations of clay to a pure polymer can substantially improve its thermal and gas permeation properties because of the planar orientations of the clay sheets. Nanocomposites possess unique properties that include thermal stability, stiffness, and low gas permeability because of their dispersion characteristics [11,12].

Previous studies [13,14] have demonstrated that a high-performance nanocomposite PI containing an organoclay can be synthesized by the thermal imidization of an aromatic precursor polymer. Even at low clay concentrations (≤10 wt %), the thermomechanical properties of these nanostructured materials can be substantially improved while decreasing their gas permeability rates. In the present study, we prepared hybrid films composed of a Co-PI and an excess of an organoclay (≤50 wt %). In contrast to hybrids with low clay concentrations, the transparency of the Co-PI hybrid with a high clay loading does diminish gradually with increasing clay concentration because of the agglomeration of clay particles. Bae et al. showed similar results in CPI nanocomposites [15]. They reported that the transmittance decreases with increasing silica concentration in CPIs synthesized with 4,4′-(hexafluoroisopropylidene) diphthalic anhydride (6FDA) and 2,2′-bis(trifluoromethyl)benzidine (TFB). These findings also suggest that the phase domains in the hybrid film are significantly larger than the wavelength of visible light [16].

To improve the thermomechanical properties of PIs, most of the CPI structures consist of rigid-rod monomers with –CF_3_ substituent in the main chain. Fluorinated PIs and Co-PIs derived from 4,4′-biphthalic anhydride (BPA) and TFB have been identified as materials that exhibit good thermomechanical properties and dimensional stabilities [17]. However, rigid rod-type CPIs have many serious problems in processing and optical application.

The introduction of a flexible monomer structure into Co-PI will significantly improve the optical transparency because of decreasing charge transfer-complexes (CT-complexes). *p*-Xylylenediamine (*p*-XDA), with its linear flexible structure, is well known to possess good gas-barrier properties and optical transparency [18]. Although the thermal stabilities and optical transparencies of these materials have been studied in detail, much less attention has been given to the dimensional stabilities and gas-barrier properties of these Co-PIs. Also, the use of *p*-XDA with an alkyl group in the main chain can increase the compatibility with organoclay and improve the physical properties of the nanocomposite. Unfortunately, there are not many studies of nanocomposites on monomers with alkyl groups in the main chain.

The main objective of the present study was to investigate the effect of the BPA concentration on the optical transparencies, and the thermal and gas permeation properties of Co-PIs composed of TFB and *p*-XDA. Co-PIs composed of a third monomer may exhibit properties that cannot be reproduced by homopolyimides. The introduction of *p*-XDA into Co-PIs that contain linear and flexible structures leads to modified optical transparencies and gas permeation behavior, which is ascribable to CT-complex formation.

This study also investigated the effects of adding a chemically modified hectorite (STN) into the Co-PIs, with a particular focus on modulating thermo-optical properties, morphologies, and gas permeabilities. Among the various Co-PI compositions, the nanocomposites were synthesized with the same composition because the molar ratio of BPA: TFB: *p*-XDA = 1:0.2:0.8 was best dispersed with organoclay. The effect of STN concentration (5–50 wt %) on the properties of a nanocomposite film was also studied, and the properties of the Co-PI nanocomposite films were also compared with those of pure Co-PI films. In most nanocomposite studies, clay is used in small amounts, usually 1~2 wt %. However, in this study, *p*-XDA containing an alkyl group having excellent compatibility with organoclay was used, so that the organoclay could be dispersed up to 50 wt %.

## 2. Experimental

### 2.1. Materials

All reagents were purchased from TCI (Tokyo, Japan). Commercially available solvents were purified by distillation, while BPA, TFB, and *p*-XDA were used as received. *N*,*N*′-dimethylacetamide (DMAc) was purified and dried over molecular sieves (4 Å) before use. Common reagents were used without further purification. The organically modified hectorite clay (STN) was obtained from the Co-op Chemical Co. (Tokyo, Japan). STN, with a cation-exchange capacity of 78 meq/100 g [19], was achieved by sifting the clay through a 325-mesh sieve to remove impurities.

### 2.2. Preparation of the Co-PI Films

The Co-PIs were synthesized using a two-step method. Poly(amic acid) (PAA) was prepared by the stoichiometric step-wise addition of BPA to dianhydride(s) in DMAc at low temperature. The monomer concentrations of the Co-PIs prepared in this study are listed in Table 1.

All samples were prepared as solutions. Because the procedures for the syntheses of the Co-PI films are very similar, only a representative example is described here, namely that for the preparation of the BPA:TFB:*p*-XDA = 1:0.2:0.8 (mole ratio) sample (sample E). TFB (2.706 g, 0.837 × 10^−2^ mol) and *p*-XDA (4.604 g, 3.347 × 10^−2^ mol) were added to 100 mL of DMAc in a 250 mL beaker, and the solution was stirred at 0 °C for 30 min to produce a homogeneous solution. BPA (12.690 g, 4.184 × 10^−2^ mol) was dissolved in 100 mL DMAc in a separate beaker and the resulting solution was added with vigorous stirring to the previously prepared DMAc solution containing TFB and *p*-XDA to give a homogeneously dispersed mixture of DMAc and PAA with a solid concentration of 9 wt %. This solution was stirred vigorously at 0 °C for 1 h, and then at 25 °C for 17 h, after which it was cast onto glass plates. 

The solvent was evaporated under vacuum at temperatures between 80 and 250 °C for a variety of reaction times. Table 2 summarizes the heat-treatment conditions employed for the preparation of each Co-PI film. The films were approximately 54–57 μm thick. No special tools were used to orient the glass plates during heat treatment because orientation can influence some of the properties of the film specimens, such as their optical transparencies, gas-permeation properties, and morphologies. The chemical structures relevant to the synthesis route are shown in Scheme 1.

### 2.3. Preparation of the Co-PI Hybrid Films

The same procedure was used for the synthesis of all BPA/TFB/*p*-XDA Co-PI/STN hybrids, irrespective of the STN concentration; consequently, only the procedure for the preparation of the Co-PI (BPA/TFB/*p*-XDA = 1:0.2:0.8 mole ratio) hybrid containing 5 wt % STN is described here. DMAc (20 mL) was mixed into to a dispersion of 0.5 g STN and 4.5 g PAA in 20 mL DMAc, after which it was stirred vigorously at room temperature for 24 h. The resulting mixture was subjected to ultrasonication followed by stirring for 5 min. This procedure was repeated six times to obtain a homogeneous composite solution. The solution was then cast onto glass plates and the solvent was evaporated under vacuum at 25 °C for 2 h followed by 50 °C for 1 h. The resulting films were then thermally treated at various temperatures to promote heterocyclization. Thermal imidization was conducted for 30 min at temperatures of 80–160 °C. Finally, the film was heated for 30 min at 190–250 °C under a steady stream of N_2_. The PI was then cooled to room temperature, and the PI film was removed from the glass plate using a hot water bath.

### 2.4. Characterization of the Co-PIs

Differential scanning calorimetry (DSC, NETZSCH F3, Berlin, Germany) and thermogravimetric analysis (TGA, TA Instruments Q500, New Castle, DE, USA) were carried out at heating rates of 20 °C/min under a flow of N_2_. The glass transition temperature (*T*_g_) was determined from the second heating using DSC. Temperature scanning was in the range of 50–350 °C in the heating cycle and from 350 to 50 °C in the cooling cycle. The coefficients of thermal expansion (CTEs) of the samples were measured using a macroexpansion probe (TA Instruments TMA-2940, New Castle, DE, USA), which was used to apply an expansion force of 0.1 N to the films at a heating rate of 5 °C/min in the 50–200 °C temperature range. 

Wide-angle X-ray diffraction (XRD) experiments were conducted at room temperature on a Rigaku (D/Max-IIIB, Tokyo, Japan) X-ray diffractometer using Ni-filtered Cu-Kα radiation. Samples were prepared for transmission-electron microscopy (TEM) by placing the STN/Co-PI hybrid films into epoxy capsules, after which they were cured at 70 °C for 24 h at reduced pressure. The cured epoxys were then microtomed (to a thickness of ~90 nm), and a layer of carbon (~3 nm) was deposited on each of the slices, after which they were placed on copper grids (200 mesh). TEM images of ultrathin sections of the STN/Co-PI hybrid films were acquired using a Leo 912 OMEGA TEM (Tokyo, Japan) at an accelerator voltage of 120 kV.

The color intensities of the polymer films were evaluated with a Minolta CM-3500d spectrophotometer at an observational angle of 10°, and a CIE–D illuminant; the CIE LAB color-difference equation was used. Ultraviolet-visible (UV-vis) spectra of the polymer films were recorded on a Shimadzu UV-3600 instrument (Tokyo, Japan). The O_2_ permeabilities of the films were measured according to ASTM E96 using a Mocon DL 100 instrument (Minneapolis, MN, USA); O_2_-transmission rates (O_2_TRs) were determined at 23 °C and at a pressure of 760 Torr.

## 3. Results and Discussion

### 3.1. Thermal Properties of the Co-PIs

The *T*_g_ of the Co-PI was observed to gradually decrease from 257 to 228 °C as the *p*-XDA molar ratio was increased from 0 to 1.0 (relative to BPA), as shown in Table 3. The presence of the *p*-XDA units in the Co-PIs results in an increase in the fractional space of the Co-PI compared to that of the material devoid of *p*-XDA, which is ascribable to the flexible structure of the methylene linkage that facilitates a greater degree of phenyl-ring rotation and a decrease in chain stiffness. 

The initial thermal decomposition temperatures (*T*_D_*^i^*) of the Co-PIs under nitrogen, which corresponds to 2% weight loss, are also listed in Table 3. Each Co-PI exhibited a single decomposition starting at a temperature of 552 °C (sample A), and their *T*_D_*^i^* values decreased linearly with increasing *p*-XDA concentration. When the types of Co-PI backbones are compared, the Co-PIs that contain flexible alkyl groups connected directly to the phenyl rings of the main chain (Scheme 1) are found to have the lowest thermal stabilities [20,21]. This conclusion is supported by the residual weight-residue data at 600 °C (*wt_R_*^600^): the Co-PI with the highest *p*-XDA concentration (sample F) exhibited the lowest *wt_R_*^600^. For example, in the Co-PI series, both *T*_D_*^i^* and *wt_R_*^600^ were observed to decrease linearly, from 552 to 481 °C and from 96% to 45%, respectively, as the *p*-XDA molar ratio was increased from 0 to 1.0 (relative to BPA) (see Table 3); the TGA thermograms of these Co-PI films are displayed in Figure 1.

The CTEs of the Co-PI films in the 50–200 °C temperature range were observed to increase significantly with the addition of *p*-XDA, as listed in Table 3. In particular, the CTEs of the Co-PI films increased from 26 to 57 ppm/°C as the molar ratio of *p*-XDA was increased from 0.2 to 1.0 (relative to BPA); these CTE values are significantly higher than those of the homopolymer (sample A) (20 ppm/°C). The PI containing *p*-XDA, but no TFB, (sample F) exhibited the highest CTE (57 ppm/°C), which is attributed to the presence of flexible alkyl linkers that prohibit close packing and chain–chain interactions. This observation indicates that the increases in the thermal expansions of these Co-PIs (sample A–F) are dependent on the dispersion of the *p*-XDA molecules and the flexible nature of the *p*-XDA unit. The *p*-XDA unit is much less rigid than the TFB moiety; therefore, it can deform or relax more readily with increasing temperature [22]. The CTE results for the Co-PI films prepared with various monomer ratios are displayed in Figure 2.

### 3.2. Co-PI Gas Permeabilities

In order to improve the permeabilities of PI films, their backbone chains must be stiffened by inhibiting their intrarotational mobilities, which improves selectivity, while the intersegmental packing of the polymer chains must be simultaneously prevented [23]. Copolymerization is one of the most useful methods for the preparation of permeable membrane materials that separate water/organic-compound mixtures because the ratio of the “hard” and “soft” segments can easily be adjusted [24]. For example, Friesen et al. [25] used a Co-PI prepared from 3,3,4,4,-benzophenonetetracarboxylic dianhydride (BTDA) and pyromellitic dianhydride (PMDA), and demonstrated that the permeation properties depended on the comonomer. 

Chain alignment increases tautness or rigidity, while decreasing the mobility within the remaining amorphous regions of the polymer [26]. By comparing TFB and *p*-XDA, we found that the methylene (-CH_2_-) groups in *p*-XDA are more linear than the bulky –CF_3_ substituents on the biphenyl rings of TFB. A higher proportion of TFB restricts conformational freedom because the bulky –CF_3_ groups hinder free rotation even along the main-chain axis, which gives rise to high torsional strain and steric hindrance. This, in turn, facilitates the easy dissociation of CF_3_ radicals, while the incorporation of the bulky –CF_3_ group inhibits close packing of the polymer chains [27,28]. Molecular chains that contain *p*-XDA methylene groups will be better aligned than those that contain TFB biphenyl linkages; as a result, the O_2_TR value of the Co-PI film decreases with increasing *p*-XDA concentration (see Table 3). For example, the O_2_TR of the Co-PI film decreased from 2313 to 98 cm^3^/m^2^/day as the *p*-XDA molar ratio was increased from 0 to 1.0 (relative to BPA).

### 3.3. Optical Transparencies of the Co-PIs

The color intensities of the Co-PI films were determined by examining the cutoff wavelengths (*λ*_o_) of their UV-vis absorption spectra. The UV-vis spectra of the Co-PI films with *p*-XDA molar ratios between 0 to 1.0 (relative to BPA) are listed in Table 4. The various Co-PI films exhibit strong absorptions with cutoffs below 400 nm, and slightly distinct peak maxima are present. The transmittance of a Co-PI film is directly related to its color intensity. These transparent Co-PI films exhibited transmittances above 85% at 500 nm and have excellent optical properties. Figure 3 displays the UV-vis spectra of the various Co-PI films recorded over the 367–391 nm range.

In general, the colorless nature of a CPI film is attributable to the strong electron-withdrawing groups of the monomeric units that inhibit CT-complex formation and decrease intermolecular interactions [29]. However, in our system, the yellow-index (YI) values were observed to decrease gradually with increasing *p*-XDA concentration (Table 4). For example, the YI decreased from 4.61 to 1.97 as the *p*-XDA molar ratio increased from 0 to 1.0 (relative to BPA). The YI value of the Co-PI film with a *p*-XDA molar ratio of 0.2 (sample B) is 4.19; however, the YI value was found to decrease from 4.19 to 1.97 as the *p*-XDA molar ratio was increased from 0.2 to 1.0 because the formation of CT-complexes in the Co-PI structures was inhibited. It seems that the flexible methylenes in the *p*-XDA units together with the TFB/BPA actively inhibits electron transfer between the electron-donating and electron-accepting units, leading to the absence of color [30,31].

The solvent-cast Co-PI films that contain *p*-XDA molar ratios between 0 to 1.0 (relative to BPA) were almost transparent, as shown in Figure 4A–F, which indicates that the addition of the *p*-XDA monomer to the Co-PI does not significantly affect its transparency. Compared to the Co-PI films with 0–0.8 molar ratios of *p*-XDA, the film devoid of TFB exhibits brighter color (see Figure 4F) because of the absence of any CT-complexes.

### 3.4. Dispersibilities of the Co-PI Hybrids

Figure 5 displays the XRD pattern of STN as well as those of 1:0.2:0.8 BPA:TFB:*p*-XDA (molar ratio) PI hybrid films containing STN loadings of between 0 and 50 wt %. The pattern of the STN surface-modified clay, exhibits a peak at a 2*θ* value of 4.60°, which corresponds to an interlayer distance (*d*) of 19.19 Å. 

As shown in Figure 5, the hybrid films with 5–50 wt % STN exhibit diffraction peaks in the 4.74–5.02° 2*θ* range (*d* = 17.58–18.62 Å), which are slightly shifted compared to that of STN and reveals that some parts of the clay have collapsed within the polyimide matrix. One hypothesis involves the thermal decomposition of the organic component of the STN that collapses the clay layers, thereby reducing the *d*-spacing. Yano et al. [32] also reported that weakly bound organic molecules detach from the clay surface; this detachment, which is caused by heat treatment during the imidization of PAA to PI, leads to a reduction in the interlayer spacing relative to that of STN [12]. Substantial increases in the intensities of the XRD peaks were observed as the clay loading increased from 5 to 50 wt %, which suggests that the organoclay agglomerates easily at higher clay loadings. 

Although XRD data are very useful for determining the *d*-spacings of ordered immiscible or intercalated polymer nanocomposites, such data may be inadequate for analyzing disordered and exfoliated materials. Therefore, the dispersion of the clay particles within the Co-PI matrix was further examined by electron microscopy, which provides information that complements those provided by the XRD experiments.

### 3.5. Morphologies of the Co-PI Hybrids

Ultramicrotomed sections of the hybrid films were subjected to TEM in order to provide direct evidence for the formation of true nanoscale composites. The TEM micrographs of the 10 wt % hybrid film, acquired at different magnifications, are shown in Figure 6. The dark lines in these images correspond to the intersections of 1-nm-thick clay layers. The TEM images at all magnifications reveal that the clay is dispersed within the polymer matrix, although some clay agglomerates with sizes greater than ~10 nm are observed, which indicates the formation of nanocomposites. These findings suggest that during the intercalative polymerization process, the clay is broken down into nanoscale building blocks that are dispersed homogeneously in the polymer matrix to form polymer–clay nanocomposites [33].

As observed for the 40 wt % STN-containing film (Figure 7), these clays are mostly agglomerated in the polymer matrix, and the peaks in the XRD patterns of these samples are attributed to the agglomerated layers (see Figure 5). Unlike the hybrids containing 10 wt % STN, the clay layers in the 40 wt % hybrid material are not intercalated in the matrix polymer, which is consistent with the XRD patterns shown in Figure 5. It is clear that agglomeration of the dispersed clay phase increases with increasing clay concentration [34]. 

### 3.6. Thermal Behavior of the Co-PI Hybrids

The thermal properties of the Co-PI hybrids with varying organoclay concentrations are summarized in Table 5. The *T*_g_ values of the Co-PI hybrid films were found to increase linearly over the 234 to 245 °C range as the clay loading was increased from 0 to 50 wt %. This increase in *T*_g_ is likely to be due to two factors: (1) the significant effect of small amounts of dispersed clay layers on the free volume of the PI, and (2) confinement of the intercalated polymer chains within the clay galleries, which prevents segmental chain motions [11]. Similar observations were made during other studies of polymer nanocomposites [35,36]. DSC thermograms of the pure Co-PI and various Co-PI hybrids are displayed in Figure 8.

Table 5 also lists the thermal stabilities of the STN hybrid films at various clay concentrations. The *T*_D_*^i^* values of the hybrid films decreased significantly, from 500 to 352 °C, as the clay concentration was increased from 0 to 50 wt %. The low values observed for the STN-containing films are ascribable to the organically modified clay and the amorphous structures of the clay interlayers that are thermally unstable. The thermal stabilities of the PI/clay-hybrid films obtained by TGA are displayed in Figure 9.

The CTE values of the STN/Co-PI hybrids in the 50–200 °C temperature range are also listed in Table 5. As was observed for *T*_g_, the CTEs of the STN hybrids improve gradually with increasing clay loading. For example, the CTE of the Co-PI hybrid decreased from 53 to 14 ppm/°C as the clay loading was increased from 0 to 50 wt %. This indicates that the magnitude of the reduction in thermal expansion due to clay layers depends on the orientations of the PI molecules and the rigid nature of the clay layers themselves. Upon heating, the in-plane-oriented PI molecules tend to relax in a direction normal to their original direction and, as a consequence, expand mainly in the out-of-plane direction [37]. The clay layers are much more rigid than the PI molecules, and do not deform or relax as easily [38]. Consequently, the clay layers effectively retard the thermal expansion of the PI molecules in the out-of-plane direction. The CTE results for the Co-PI films with various clay concentrations are displayed in Figure 10.

### 3.7. Gas Permeabilities of the Co-PI Hybrids

The permeability of a hybrid depends on several factors, including the quantity, length, and width of the clay particles, as well as their orientation and dispersion [39]. Numerous studies [40,41] have shown that the aspect ratios of exfoliated clay particles play critical roles in controlling the microstructures of polymer/clay nanocomposites and their gas barrier properties. Although polymer/clay nanocomposites are known to exhibit gas barrier properties that are superior to those of conventional composite systems, the dependence of these properties on factors such as the orientations of the matrix sheets and the extent of clay aggregation and dispersion (intercalation, exfoliation, or intermediate structures) is not well understood. Bharadwaj [39] constructed a model for the barrier properties in polymer-layered silicate nanocomposites based only on the tortuosity arguments described by Nielsen [42]. Correlations between sheet length, concentration, relative orientation, and aggregation state are expected to provide guidance for the design of better barrier materials using the nanocomposite approach.

The introduction of the inorganic material into the polymer film affects its gas permeability. In general, the gas permeabilities of hybrid films are lower than those of polymer films and are independent of gas type. This behavior is attributed to the high aspect ratios and rigidities of the inorganic clay platelets in the polymer matrix [43]. To further characterize the barrier properties of the PI/clay hybrids fabricated by the intercalation of polymer chains in the STN galleries, we evaluated the permeabilities of the resultant Co-PI hybrids to O_2_.

The O_2_TRs of the hybrid films with clay concentration of 0–50 wt % are summarized in Table 5. Because this article explores the performance of barrier films containing aligned impermeable clay particles, we discuss our results in terms of the relative permeability, *P_c_*/*P_p_*, where *P_p_* is the permeability of the pure polymer, and *P_c_* is the permeability of the composite. The relative permeability of the 40 wt % Co-PI hybrid film was the lowest (<10^−2^ cm^3^/m^2^/day), and much lower than that of the pure Co-PI film (223 cm^3^/m^2^/day). This reduced permeability is ascribable to the presence of layers of dispersed large-aspect-ratio clay in the polymer matrix, as has previously been shown for other hybrid films [44].

The presence of clay introduces a tortuous path for penetrant diffusion. The reduced permeability arises from the longer diffusive path that the penetrants must travel in the presence of the clay filler [23]. A sheet-like morphology is particularly efficient for maximizing this path length because of the large length-to-width (L/W) ratio compared to other filler shapes, such as spheres or cubes. The average L/W value of the STN was ~146. However, an increase in oxygen permeability, to 10 cm^3^/m^2^/day, was observed at an organoclay concentration of 50 wt %. This increase in permeability is attributable to the agglomeration of the clay particles above a critical clay concentration [13,45]. 

Overall, these results show that clay in the Co-PI increases the tortuous-path length travelled by the gas molecules and the interactions between O_2_ and the organically modified clay molecules. Furthermore, films containing higher amounts of clay, up to 40 wt %, appear to be much more rigid, which results in decreases in their gas permeabilities.

### 3.8. Optical Transparencies of the Hybrids

The color intensities of the hybrid films were determined by measuring *λ*_0_ in their UV-vis absorption spectra. The color intensities of the Co-PI hybrid films with various clay concentration are listed in Table 6. The *λ*_0_ value of the Co-PI hybrid was virtually unaffected by the clay loading; specifically, it increased from 378 to 383 nm as the STN concentration was increased from 0 to 50 wt %. The UV-vis absorption spectra of the Co-PI hybrid films with varying amounts of clay are displayed in Figure 11.

The transmittances of the Co-PI hybrid films decreased with increasing clay concentration due to clay-particle agglomeration [14,44,45]. The transmittance at 500 nm decreased significantly, from 86 to 74%, as the STN concentration was increased from 0 to 50 wt % (Table 6); clay agglomeration was evidenced by XRD and TEM (Figure 5, Figure 6 and Figure 7). These STN hybrid films exhibited maximum UV transmittances of ≥74% at 500 nm.

Table 6 also reveals that the color intensities of the Co-PI hybrid films are affected by clay concentration. The Co-PI hybrid films with lower clay concentrations had lower YI values than the corresponding hybrids with higher clay concentration. The YI value of the Co-PI hybrid with 0 wt % STN (pure Co-PI) was 2.93. When the clay loading was first increased to 5 wt % and then to 50 wt %, significant increases in the YI value, to 4.77 and then to 19.26, respectively, were observed, which is due to clay-particle agglomeration. This increase in the YI suggests that clay particles are better dispersed in the polymer matrix at lower clay loadings [13].

All solvent-cast hybrid films with clay concentration in the 0–10 wt % range were almost transparent and colorless, as shown in Figure 12. However, their optical transparencies decreased slightly as the clay concentration was increased above this range. The clay hybrid film containing 20 wt % clay was slightly cloudier and more yellowish than films containing 5–10 wt % clay; however, its optical transparency was still good as there were no problems in visualizing images through these films. The levels of transparency were not significantly affected as the clay concentration was increased from 5 to 50 wt %, as shown in Figure 12, and there were only small differences between the optical transparencies of the various samples. 

## 4. Conclusions

Co-PIs were synthesized from BPA and two diamine monomers, namely TFB and *p*-XDA, both of which are commonly used in colorless and transparent PIs. The Co-PI films based on BPA with various *p*-XDA mole ratios were systematically investigated. The thermal properties of the Co-PI films were found to deteriorate with increasing *p*-XDA concentration, however, the O_2_TRs and YIs of the Co-PI films improved with increasing *p*-XDA concentration. The observed differences in these properties are related to the chemical structure of *p*-XDA in the Co-PI main chain. 

Co-PI hybrid films (BPA:TFB:*p*-XDA = 1:0.2:0.8 molar ratio) were also prepared by the solution intercalation method using STN as an organoclay. The *T*_g_ and CTEs of the Co-PI hybrids were observed to improve with increasing clay loadings of up to 50 wt %. However, up to a critical loading, the O_2_TRs of the Co-PI hybrid films decreased with the addition of the STN, and then increased above that critical concentration. Morphological studies indicated that the clay was well dispersed in the Co-PI matrix without significant agglomeration of particles in hybrids with lower clay loadings. In contrast, particle agglomeration was observed in hybrids with higher clay loadings (50 wt %), resulting in inferior O_2_TR. The optical transparencies of the Co-PI hybrid films also gradually decreased with increasing STN concentration (5–50 wt %), which is ascribable to the agglomeration of clay particles.

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
