# Peer review of "Colorless and Transparent Copolyimides and Their Nanocomposites: Thermo-Optical Properties, Morphologies, and Gas Permeabilities"

_polymers, 2019, doi:10.3390/polym11040585_

Round 1

Reviewer 1 Report

polymers-452703 developed a series of linear aromatic copolyimides (Co-PIs) by reacting 4,4′-biphthalic anhydride (BPA) with various molar contents of 2,2′-bis(trifluoromethyl)benzidine (TFB) and p-xylylenediamine (p-XDA). Effects of molar ratios of p-XDA on properties like thermal, gas permeability, optical transparencies etc. were investigated. Co-PI hybrid films were also prepared using different loadings of organically modified hectorite. Dispersibility, morphology and glass transition temperature etc. of the hybrid films were reported. Generally, the starting compounds of this manuscript are not new, and no outstanding property was found to be associated with the prepared films. Simply investigating the effects of compositions on properties of polymers may not be novel enough to guarantee this manuscript to be published in Polymers. Thus, I recommend this manuscript to be rejected. 

Author Response

Dear Editor

This is my response to your comments regarding our paper “Colorless and Transparent Copolyimides and Their Nanocomposites: Thermo-optical Properties, Morphologies, and Gas Permeabilities

(polymers-452703) in polymers.

Thank you very much for the referee's comments. We have carefully revised the manuscript following the comments of the referee.

Response to Reviewer-1 comments: The supplement was marked in red in the manuscript.

Point-1: Indication of originality

Response-1: As the referee pointed out, we explained the originality of this paper and the properties that can not be seen in other papers.

“To improve the thermo-mechanical properties of PIs, most of the CPI structures consist of rigid-rod monomers with -CF3 substituent in the main chain. Fluorinated PIs and Co-PIs derived from 4,4′-biphthalic anhydride (BPA) and TFB have been identified as materials that exhibit good thermo-mechanical properties and dimensional stabilities [17]. However, rigid-rod type CPIs have many serious problems in processing and optical application. 

The introduction of a flexible monomer structure into Co-PI will significantly improve the optical transparency because of decreasing charge transfer-complexes (CT-complexes). p-Xylylenediamine (p-XDA), with its linear flexible structure, is well known to possess good gas-barrier properties and optical transparency [18]. Although the thermal stabilities and optical transparencies of these materials have been studied in detail, much less attention has been given to the dimensional stabilities and gas-barrier properties of these Co-PIs. Also, the use of p-XDA with an alkyl group in the main chain can increase the compatibility with organoclay and improve the physical properties of the nanocomposite. Unfortunately, there are not many studies of nanocomposites on monomers with alkyl groups in the main chain.was added in Introduction (see Page2). 

I hope this revision is satisfactory for your further process. 

Best,

Jin-Hae Chang

Professor

Reviewer 2 Report

- - Page 2 line 55-56:  “In contrast to hybrids with low clay concentrations, the transparency of the Co-PI hybrid  with a high clay loading does diminish gradually with increasing clay concentration because of the agglomeration of clay particles”

It is a widely known fact that even in the case of a nanofiller well dispersed in the polymer matrix, the decrease in transparency of materials can be observed. The authors should have taken note of that in their work.

- The range of the temperature during Differential scanning calorimetry should be indicated in the methodology part of the work.

- It should be explained why in the case of pristine STN the intensity on the XRD curve is lower than in the case of composites. (figure 5) page 9.

- It should be mentioned in the manuscript that reduction in the interlayer spacing can indicate that conventional composites were obtained.

- The initial weight loss temperature (table 5) is not consistent with results presented in figure 9. In figure 9 it can clearly be seen that thermal stability significantly decreases in relation to the increase in STH content in the studied materials.

- Summarizing: The conclusions are not consistent with the results presented in the work: thermal stability of the obtained composites decreased and the XRD and TEM results indicate that conventional composites were obtained. Moreover the Authors do not explain why composition Co-PI (1:02:08) had been chosen in order to obtain the nanocomposites. The reason for using such a high clay content should also be explained.

I suggest a minor revision of the manuscript.

Author Response

Dear Editor

This is my response to your comments regarding our paper “Colorless and Transparent Copolyimides and Their Nanocomposites: Thermo-optical Properties, Morphologies, and Gas Permeabilities

(polymers-452703) in polymers.

Thank you very much for the referee's comments. We have carefully revised the manuscript following the comments of the referee.

Response to Reviewer-2 comments: The supplement was marked in red in the manuscript.

Point-1: It is a widely known fact that even in the case of a nanofiller well dispersed in the polymer matrix, the decrease in transparency of materials can be observed. The authors should have taken note of that in their work.

Response-1: As the referee pointed out, the results of other researchers have been described as an example. In Page 2, “Bae et al. showed similar results in CPI nanocomposites [15]. They reported that the transmittance decreases with increasing silica concentration in CPIs synthesized with 4,4′-(hexafluoroisopropylidene) diphthalic anhydride (6FDA) and 2,2′-bis(trifluoromethyl)benzidine (TFB).” was added.   

Point-2: The range of the temperature during Differential scanning calorimetry should be indicated in the methodology part of the work.

Response-2: As the referee pointed out, The glass-transition temperature (Tg) was determined from the second heating using DSC. Temperature scanning was in the range of 50-350 °C in the heating cycle and from 350 to 50 °C in the cooling cycle.” was added in page 5, section 2.4. Characterization of the Co-PIs

Point-3: It should be explained why in the case of pristine STN the intensity on the XRD curve is lower than in the case of composites. (figure 5) page 9.

Response-3: In the process of obtaining Figure 5, there was a big mistake. As the referee pointed out, Figure 5 was redrawn considering the intensity.

Point-4: It should be mentioned in the manuscript that reduction in the interlayer spacing can indicate that conventional composites were obtained.

Response-4: As the referee pointed out, Yano et al. [32] also reported that weakly bound organic molecules detach from the clay surface; this detachment, which is caused by heat treatment during the imidization of PAA to PI, leads to a reduction in the interlayer spacing relative to that of STN [12].” Was supplemented in section 3.4. Dispersibilities of the Co-PI hybrids

Point-5: The initial weight loss temperature (table 5) is not consistent with results presented in figure 9. In figure 9 it can clearly be seen that thermal stability significantly decreases in relation to the increase in STH content in the studied materials.

Response-5: The results in Table 5 and Figure 9 are in good agreement. As the amount of STN increases in the nanocomposites, the initial decomposition temperature is steadily decreasing.

Point-6 in Summarizing: Authors do not explain why composition Co-PI (1:02:08) had been chosen in order to obtain the nanocomposites. The reason for using such a high clay content should also be explained.

Response-6: As the referee pointed out, “Among the various Co-PI compositions, the nanocomposites were synthesized with the same composition because the molar ratio of BPA: TFB: p-XDA = 1: 0.2: 0.8 was best dispersed with organoclay. The effect of STN concentration (5–50 wt%) on the properties of a nanocomposite film was also studied, and the properties of the Co-PI nanocomposite films were also compared with those of pure Co-PI films. In most nanocomposite studies, clay is used in small amounts, usually 1 ~ 2 wt%. However, in this study, p-XDA containing an alkyl group having excellent compatibility with organoclay was used, so that the organoclay could be dispersed up to 50 wt%.” was supplemented in Page 3, the end of Introduction section.

I hope this revision is satisfactory for your further process. 

Best,

Jin-Hae Chang

Professor

Reviewer 3 Report

In this manuscript by Kwark et al. the authors have described a method of formation of transparent materials and characterized a range of their properties (thermal, microstructure, gas permeability, etc.). The work is timely, important and carried out very well. There is no reason not to accept this publication as is. 

Author Response

Dear Reviewer-3

Many thanks for accepting the paper.

Best,

Jin-Hae Chang

Round 2

Reviewer 1 Report

Thanks for the authors' responses. I have no more concern about this manuscript.